# LSGP-USFNet: Automated Attention Deficit Hyperactivity Disorder Detection Using Locations of Sophie Germain’s Primes on Ulam’s Spiral-Based Features with Electroencephalogram Signals

**DOI:** 10.3390/s23167032

**Published:** 2023-08-08

**Authors:** Orhan Atila, Erkan Deniz, Ali Ari, Abdulkadir Sengur, Subrata Chakraborty, Prabal Datta Barua, U. Rajendra Acharya

**Affiliations:** 1Electrical-Electronics Engineering Department, Technology Faculty, Firat University, 23119 Elazig, Turkey; oatila@firat.edu.tr (O.A.); edeniz@firat.edu.tr (E.D.); ksengur@firat.edu.tr (A.S.); 2Computer Engineering Department, Engineering Faculty, Inonu University, 44280 Malatya, Turkey; ali.ari@inonu.edu.tr; 3Faculty of Science, Agriculture, Business and Law, University of New England, Armidale, NSW 2351, Australia; 4Faculty of Engineering and Information Technology, University of Technology Sydney, Ultimo, NSW 2007, Australia; 5School of Information Systems, University of Southern Queensland, Springfield, QLD 4300, Australia; 6School of Mathematics, Physics and Computing, University of Southern Queensland, Springfield, QLD 4300, Australia; rajendra.acharya@usq.edu.au

**Keywords:** ADHD detection, EEG signals, Ulam’s spiral, Sophie Germain’s primes, SVM

## Abstract

Anxiety, learning disabilities, and depression are the symptoms of attention deficit hyperactivity disorder (ADHD), an isogenous pattern of hyperactivity, impulsivity, and inattention. For the early diagnosis of ADHD, electroencephalogram (EEG) signals are widely used. However, the direct analysis of an EEG is highly challenging as it is time-consuming, nonlinear, and nonstationary in nature. Thus, in this paper, a novel approach (LSGP-USFNet) is developed based on the patterns obtained from Ulam’s spiral and Sophia Germain’s prime numbers. The EEG signals are initially filtered to remove the noise and segmented with a non-overlapping sliding window of a length of 512 samples. Then, a time–frequency analysis approach, namely continuous wavelet transform, is applied to each channel of the segmented EEG signal to interpret it in the time and frequency domain. The obtained time–frequency representation is saved as a time–frequency image, and a non-overlapping *n* × *n* sliding window is applied to this image for patch extraction. An *n* × *n* Ulam’s spiral is localized on each patch, and the gray levels are acquired from this patch as features where Sophie Germain’s primes are located in Ulam’s spiral. All gray tones from all patches are concatenated to construct the features for ADHD and normal classes. A gray tone selection algorithm, namely ReliefF, is employed on the representative features to acquire the final most important gray tones. The support vector machine classifier is used with a 10-fold cross-validation criteria. Our proposed approach, LSGP-USFNet, was developed using a publicly available dataset and obtained an accuracy of 97.46% in detecting ADHD automatically. Our generated model is ready to be validated using a bigger database and it can also be used to detect other children’s neurological disorders.

## 1. Introduction

Attention deficit hyperactivity disorder (ADHD) is a disease defined as a neurological mental disorder [1]. This disease usually manifests itself in childhood in the form of inattention, hyperactivity, forgetfulness, and uncontrolled or immediate and impulsive reactions [2]. Because mental disorders are linked to brain function, researchers often use electroencephalogram (EEG) signals to diagnose mental disorders [3]. EEG signals are complex and nonlinear continuous signals. Therefore, it is very difficult to diagnose by visually examining the EEG signals of ADHD and healthy children [4]. In addition, if the decision-making mechanism is operated using existing machine learning applications, it does not provide precise information about the accuracy of the diagnosis [5]. Up to now, there have been various machine-learning-based approaches employed for ADHD detection using EEG signals [6,7,8,9,10,11,12,13,14,15,16,17]. Table 1 summarizes the works conducted on the automated detection of ADHD using EEG signals.

It can be seen from Table 1 that most of the methods have used their own datasets and are complex methods. The degree of flexibility offered by the nonlinear feature extraction approaches used directly on EEG data may not be sufficient for feature extraction in multicomponent EEG signals. The Fourier transform-based techniques are unable to localize an event precisely in the time–frequency domain, and labor-intensive filtering-based procedures are needed for accurate filter coefficient adjustment for boundaries with sharp edges. The techniques for automated feature extraction and classification like long short-term memory (LSTM) and convolutional neural network (CNN) are complex and computationally intensive [18,19]. Choosing a wavelet type is necessary for wavelet-based analysis, but mode mixing affects empirical mode decomposition (EMD). The shortcomings of the earlier techniques lead to limited success in terms of assessment measures. Thus, in this study, a novel strategy based on gray-level patterns discovered using Sophia Germain primes and Ulam’s spiral is proposed for accurate ADHD detection. The EEG signals are first filtered to remove noise, and then they are segmented using a non-overlapping sliding window containing 512 samples of data. The segmented EEG signal is then subjected to continuous wavelet transform as a time–frequency analysis technique to understand it in the time and frequency domain. A non-overlapping *n* × *n* sliding window is used for the time–frequency picture created from the acquired time–frequency representation to extract a patch. On these patches, an *n* × *n* Ulam’s spiral is localized, and the gray tones where the fitting Sophie Germain’s primes are found are obtained. For the creation of the representative features for the ADHD and normal classes, all gray tones from all patches are concatenated. Based on the representative characteristics, the ReliefF gray tone selection method is used to obtain the final, most significant gray tones. The obtained final feature vectors are classified with an SVM classifier to detect ADHD automatically.

The novelties of this study are as follows:

To the best of our knowledge, this is the first work to use gray-level patterns employing Sophia Germain primes and Ulam’s spiral method for accurate automated ADHD detection.

The obtained results are robust and accurate as we have employed ten-fold cross-validation.

## 2. Materials and Methods

In this section, the components of the proposed method, namely the dataset, developed LSGP-USFNet, details of Ulam’s spiral and Sophie Germain’s primes-based feature extraction, and feature selection and classification techniques, are described. The step-wise illustrations of the proposed methodology are given in Figure 1, Figure 2, Figure 3 and Figure 4.

### 2.1. Dataset

The dataset used in this study was taken from the IEEE dataport [20]. Subjects in the dataset were aged from 7 to 12 years old. According to DSM-IV criteria, an expert psychiatrist diagnosed ADHD in patients who used Ritalin for up to six months [21]. In the normal group, there were no children with a history of drug misuse, epilepsy, high-risk behaviors, or mental illnesses. The respondents were instructed to count the characters from a collection of photographs as part of a visual attention assignment. Each large-sized picture has between 5 and 16 characters, all of which must be legible and easy to count. After the child responded to the continuous stimulation of EEG recording, each image was shown to the subject without interruption. The child’s success throughout this cognitive visual activity determines how long each EEG signal lasts. Using 10–20 EEG recording devices, the signals were obtained from the occipital, central, parietal, and frontal brain areas (Fz, Cz, P3, P4, T5, T6, Pz, F3, F4, C3, T3, C4, T4, F7, F8, O1, O2, Fp1, and Fp2). Two electrodes were positioned above and below the right eye to −The normal and ADHD EEG signals from the dataset are shown in Figure 1.

### 2.2. Proposed Model

Analyzing EEG signals in their original state is challenging because of their complex nature. Research has revealed that dividing EEG signals into distinct frequency patterns has yielded valuable information about differentiating ADHD subjects from normal ones. However, the classification performance of such a system has been negatively affected by drawbacks, such as limited precision in time–frequency localization, inadequate noise reduction, difficulties in parameter adjustment, absence of mathematical modeling, and an inability to rectify backward errors. Thus, in this paper, a novel workflow is introduced to mitigate the weaknesses of the previous methods [1,2,3,4,5,6,7,8,9,10,11,12,13,14,15,16,17].

In the proposed LSGP-USFNet approach, the EEG recordings of each individual are initially stratified into a 4 s epoch with 512 samples after filtering the EEG signals [8]. During filtering, a notch filter operating at a frequency of 50 Hz is employed to effectively eliminate power line disturbances and undesirable noise components. Concurrently, a sixth-order Butterworth filter with a passband ranging from 0 to 60 Hz is employed to mitigate the presence of additional unwanted artifacts within the signal. From the entire dataset, 1843 segments of normal patients and 2330 segments of ADHD subjects were acquired. For each channel of the EEG segments, the CWT is applied to convert them from a one-dimensional (1D) signal to a time–frequency representation. The CWT is computed using the analytic Morse wavelet characterized by specific parameters: a symmetry parameter (gamma) set to 3 and a time–bandwidth product of 60. For this transformation, 10 voices per octave are utilized to adequately capture the frequency content of the signal. Using the wavelet energy distribution in the frequency and time domain, the minimum and maximum scales are carried out automatically. Figure 2 shows the EEG signal segmentation and time–frequency representation of the EEG signal.

Ulam’s spiral constitutes a visual representation portraying the natural numbers arranged in a spiral configuration, commencing from the center and extending outward in a counter-clockwise manner. Upon plotting the natural numbers onto Ulam’s spiral, intriguing patterns manifest, particularly pertaining to prime numbers. Notably, an observable pattern is that prime numbers align along distinct diagonal alignments on the spiral. The presence of these distinctive diagonal prime patterns on Ulam’s spiral represents a visual phenomenon, evoking considerable interest among mathematicians and clinicians. Nevertheless, the precise mathematical explanation underlying the emergence of these patterns constitutes an active and ongoing area of research among clinicians. These patterns are able to capture subtle changes in the EEG signals of ADHD and normal classes and yield high classification performances like molecular patterns used [22].

### 2.3. Continuous Wavelet Transform

The continuous wavelet transform (CWT) is a mathematical technique used for analyzing signals in both time and frequency domains [23]. It provides a powerful tool for detecting localized features or patterns within a signal that may vary in scale. The CWT is particularly useful in applications such as signal processing, image analysis, and time–frequency analysis. The CWT uses wavelet functions that are oscillatory in nature and possess localized characteristics in both time and frequency domains. These wavelets are derived from a fundamental function known as the mother wavelet, denoted by *ψ(t)*. The CWT involves convolving the input signal, *f(t)*, with a set of wavelets, which are dilated and translated versions of the mother wavelet. Mathematically, the CWT can be expressed as
(1)$$X\left(a,b\right)=\int f\left(t\right)*\Psi^{*}(\frac{t-b}{a})\partial t$$

In this equation, *a* and *b* represent the scale and translation parameters, respectively. The asterisk (*) denotes the complex conjugate of the wavelet function. The integral *(∫)* is taken over the entire time domain, integrating the product of the signal and the wavelet function over time. This process is performed for different values of *a* and *b*, which represent different scales and translations. While the scale parameter, *a*, controls the width of the wavelet function, the translation parameter, *b*, shifts the wavelet along the time axis. The CWT coefficients provide information about the presence and strength of different frequency components at different scales in the input signal [24,25]. The result of the convolution operation is a representation of the input signal in the time–frequency plane, providing information about both the temporal and spectral characteristics of the signal at different scales. The time–frequency representations are saved as color images, as shown in Figure 2.

### 2.4. Feature Extraction Based on the Locations of Sophie Germain’s Primes on Ulam’s Spiral

After time–frequency images of the segmented EEG channels are constructed, a patch division operation is employed on time–frequency images to divide them into non-overlapping square regions. Patch dividing plays a crucial role in image classification by allowing for localized analysis, enhancing feature extraction, providing translation invariance, mitigating memory and computational constraints, and handling scale variations [26]. It enables machine learning models to capture detailed information from different regions of an image and improves the overall performance of the classification task. The non-overlapping patch division is illustrated in Figure 3. In Figure 3, the input time–frequency image, which is 225 × 225 in size, is divided into 15 × 15 image patches using a non-overlapping 15 × 15 sliding window. Thus, a total of 225 patch images are obtained for the subsequent feature extraction process.

#### 2.4.1. Ulam’s Spiral

As shown in Figure 3, a 15 × 15 Ulam’s spiral is constructed and aligned on an image patch. Ulam’s spiral, named after the mathematician Stanislaw Ulam, is a visualization technique that displays prime numbers in a unique and visually appealing way [27]. It involves plotting the natural numbers in a spiral pattern and highlighting the prime numbers within the spiral. To construct Ulam’s spiral, the natural numbers are arranged in a square grid, starting from the center and spiraling outward in a counter-clockwise direction. The numbers are placed in a spiral pattern, with each subsequent number being placed adjacent to the previous number.

Once the spiral is constructed, the prime numbers within the spiral are typically highlighted by shading or coloring them differently from the composite numbers. Prime numbers, which are only divisible by one and themselves, appear as isolated points within the spiral, forming intriguing patterns and clusters. Ulam’s spiral provides a visual representation of the distribution and patterns of prime numbers. It reveals fascinating characteristics such as diagonal lines of primes and the apparent absence of a predictable pattern or structure. The visualization can help researchers and enthusiasts explore and study the properties of prime numbers, as well as inspire curiosity and interest in number theory.

#### 2.4.2. Sophie Germain’s Prime Numbers

As Ulam’s spiral is aligned on the image patch, Sophie Germain’s primes, which are located on Ulam’s spiral, are used to fuse the gray levels of the time–frequency patch images. Sophie Germain’s primes, named after the French mathematician Sophie Germain, are a special class of prime numbers that have important applications in number theory and cryptography [28]. They are closely related to the concept of safe primes and provide a foundation for various mathematical algorithms and encryption protocols. A prime number, p, is considered a Sophie Germain’s prime if 2p + 1 is also a prime number. In other words, if p is a Sophie Germain’s prime, then 2p + 1 is a prime as well. Sophie Germain’s primes have significant applications in cryptography, specifically in the field of public-key encryption. They serve as the foundation for protocols like the Diffie–Hellman key exchange and the Rivest–Shamir–Adleman (RSA) algorithm. These algorithms rely on the difficulty of factoring large numbers, which is enhanced by the use of Sophie Germain’s primes.

### 2.5. Feature Selection and Classification

In this subsection, the feature reduction and feature classification are carried out. Figure 4 gives an illustration of these procedures. After the feature extraction, which is introduced in the previous sections, for a sample input image, a 64,125-dimensional feature vector is obtained, making the classification stage time-consuming. To reduce the number of features while keeping their discriminating property, a feature selection mechanism is employed. The efficient relief feature selection algorithm is considered for feature selection, and the SVM method is used for classification. In the next subsections, they are briefly explained.

#### 2.5.1. Relief Feature Selection

ReliefF (relief feature selection) is a feature selection algorithm that aims to identify relevant features from a given dataset [29]. It is particularly useful for classification problems and has been widely applied in machine learning and data mining tasks. The main idea behind ReliefF is to estimate the quality or importance of features based on how well they discriminate between instances of different classes. It takes into account both the local and global characteristics of the data.

The ReliefF algorithm is briefly presented below:1.Initialization: initialize the weight vectors for each feature to zero.2.For each instance in the dataset:2.1.Randomly select another instance from the same class (nearest hit).2.2.Randomly select an instance from a different class (nearest miss).3.Update the weight vectors:3.1Increase the weight of features that are similar for the instance and its nearest miss.3.2.Decrease the weight of features that are similar for the instance and its nearest hit.4.Repeat steps 2 and 3 for a fixed number of iterations or until convergence.5.Calculate the final feature scores based on the accumulated weights.6.Select the top-k features with the highest scores as the relevant features.

ReliefF considers the differences in feature values between instances and their nearest hits and misses. Updating the weights based on these differences focuses on features that have a higher impact in distinguishing between different classes. The feature scores obtained from ReliefF indicate the relevance or importance of each feature in the classification task. Features with higher scores are considered more informative and are selected for further analysis or model building. ReliefF offers several advantages, such as its ability to handle both categorical and continuous features, as well as its robustness to noise and irrelevant features. It also has a low computational complexity compared to some other feature selection methods.

#### 2.5.2. SVM Classifier

The support vector machine (SVM) algorithm is a supervised machine learning algorithm used for classification and regression tasks [30]. In classification, SVM aims to find an optimal hyperplane that separates data points of different classes, while in regression, it finds a hyperplane that best fits the data [31]. Given a training dataset with labeled examples (*x_i_, y_i_*), where x_i_ represents the input features and *y_i_* represents the corresponding class labels (+1 or −1), the goal is to find a hyperplane defined by the following equation:(2)$$w^{T}x+b=0$$
where *w* is the normal vector to the hyperplane and *b* is the bias term. The hyperplane divides the feature space into two regions, each representing one class. The objective of SVM is to maximize the margin, which is the distance between the hyperplane and the closest points from each class. These closest points, known as support vectors, determine the position and orientation of the hyperplane. SVM aims to find the hyperplane that maximizes this margin, thus achieving the best separation between the classes. Mathematically, the optimization problem for SVM can be formulated as
(3)$$minimize\;\frac{1}{2}{\left\|w\right\|}^{2}+C\sum \xi_{i}$$
(4)$$subject\;to\;y_{i}\left(w^{T}+b\right)\geq 1-\xi_{i}\;and\;\xi_{i}\;\geq 0$$
where ||*w*|| represents the Euclidean norm of the weight vector *w*, *C* is a hyperparameter controlling the trade-off between margin maximization and training error minimization, and ξ*_i_* are slack variables that allow for soft-margin classification, allowing some misclassification. To handle nonlinearly separable data, SVM employs the kernel trick, which implicitly maps the input features into a higher-dimensional feature space. This allows the SVM to find a linear hyperplane in the transformed feature space, which corresponds to a nonlinear decision boundary in the original feature space.

## 3. Experimental Results

The proposed LSGP-USFNet approach used a 512-sample length sliding window to segment the input EEG signals. Thus, a total of 1843 segments of normal subjects and 2330 segments of ADHD subjects were acquired. A 10-fold cross-validation data division technique was used in the experiments. Performance measures, namely classification accuracy, sensitivity, precision, and F1 score evaluation metrics, were used. The SVM parameters were adjusted using a hyperparameter search algorithm [32]. And the number of nearest neighbors and the number of selected features using the ReliefF algorithm were set to 5 and 7000. These values were determined during the experiment. The dataset consists of two parts, where in the first one (Part 1), there are 30 ADHD and 30 normal subjects, and in the second part (Part 2), there are 31 ADHD and 30 normal subjects. After the segmentation of the EEG signals, in the first part, there are 1198 ADHD samples and 848 normal samples. And in the second part, there are 1132 samples for ADHD and 995 samples for the normal subjects. Table 2, Table 3, Table 4, Table 5 and Table 6 show the average classification evaluation metrics for various sizes of Ulam’s spirals, such as 9 × 9, 15 × 15, 25 × 25, 45 × 45, and 75 × 75.

In Table 2, the results for a patch size of 9 × 9 are given. The rows of Table 2 indicate the data for Part 1, Part 2, and Part 1 + Part 2, and the columns show the average evaluation metrics of accuracy, sensitivity, precision, and F1 scores. As seen in Table 2, the best average classification accuracy score with a 10-fold cross-validation technique of 98.78 ± 0.0070% was obtained for the Part 1 dataset. Similarly, the best average sensitivity (98.24 ± 0.0081%) and F1 score (98.53 ± 0.0085%) values were obtained for the Part 1 dataset. The best average precision score of 98.57 ± 0.0096% was obtained for the Part 2 dataset. The lowest average evaluation metrics were obtained for the combination of Part 1 and Part 2 datasets, as shown in the fourth row of Table 2.

In Table 3, the evaluation results with a 15 × 15 patch size are given. It can be noted from Table 3 that the best average accuracy, sensitivity, precision, and F1 score of 98.97 ± 0.0084%, 99.06 ± 0.0122%, 98.49 ± 0.0088%, and 98.77 ± 0.0101%, respectively, were obtained. The second-best average evaluation scores were also obtained for the Part 2 dataset, and the lowest scores were obtained for the Part 1 + Part 2 dataset.

In Table 4, a comparably larger patch size of 25 × 25 was used to study its effect on the performance of the proposed method. As seen in Table 4, the obtained results for the Part 1 dataset were better than those for the Part 2 and Part 1 + Part 2 datasets. A 98.25 ± 0.0025% average accuracy score, 98.71 ± 0.0067% average sensitivity, 98.16 ± 0.0043% precision, and a 98.44 ± 0.0027% F1 score were obtained for the Part 1 dataset. Similar to the previous patch sizes, for a patch size of 25 × 25, the achievements for Part 2 were better than those for the Part 1 + Part 2 dataset and lower than those for the Part 1 dataset.

In Table 5 and Table 6, larger patch sizes, such as 45 × 45 and 75 × 75, were used to study the effect of the larger patch sizes on ADHD detection. It can be noted from these tables (Table 5 and Table 6) that increasing the patch size did not increase the performance of the proposed method. Conversely, the performance of the proposed method decreased dramatically when larger patch sizes were considered. But it is worth mentioning that for a patch size of 45 × 45, the performance metrics for the Part 2 dataset were better than those for the Part 1 and Part 1 + Part 2 datasets. It can be noted from Table 2, Table 3, Table 4, Table 5 and Table 6 that, only for the 45 × 45 patch size, the obtained evaluation metrics for the Part 2 dataset were better than those for the Part 1 dataset.

We did not apply patch extraction to the input images. In other words, the input time–frequency images were used to acquire the features to study the effect of patch extraction on the performance of the proposed method. As seen in Table 7, the obtained evaluation metrics were low, and when these evaluation results were compared with the results given in Table 2, Table 3, Table 4, Table 5 and Table 6, it was observed that patch extraction increased the performance of the proposed method positively.

In Table 8, besides the results obtained using the ReliefF technique for the Part 1 dataset, the performances of the other feature selection methods such as neighborhood component analysis (NCA) [33], Pearson correlation coefficient (PCC) [34], and term variance feature selection (TVFS) [35] are also given. It may be noted from the table that average accuracies of 95.46 ± 0.0175%, 94.63 ± 0.0232%, and 94.38 ± 0.0103% were obtained for the NCA, PCC, and TVFS techniques, respectively. NCA obtained better achievement than the PCC and TVFS techniques. And ReliefF yielded the best average evaluation metrics for this experiment.

An experiment was also carried out with ReliefF feature selection using the Part 1 + Part 2 dataset to determine the number of features obtained using the ReliefF technique. To this end, the number of features was initiated at 1000 and increased to 14,000 with increments of 1000. The graph of average accuracy (%) versus the number of features obtained is given in Figure 5. As seen in Figure 5, the best average accuracy of 97.46% was obtained with 7000 features using the ReliefF technique.

### Ablation Study

In the proposed approach, there are two independent parts, namely the feature extraction part and the feature selection and classification part. In the first part, various sizes of patch division windows were used, such as 9 × 9, 15 × 15, 25 × 25, 45 × 45, and 75 × 75. And for the second part, ReliefF-based feature selection and SVM-based feature classification were carried out. In Table 9, the running times in seconds for various window-size-based feature extraction are given for the datasets.

It can be noted from Table 9 that the running times obtained for various window sizes of the patches during feature extraction are different. For the 9 × 9 size of the window, the running times were 1431 and 1260 s for the Part 1 and Part 2 datasets, respectively, and for the 75 × 75 size of the window, the running times were 940 and 984 s for the Part 1 and Part 2 datasets, respectively.

## 4. Discussions

In this paper, a novel approach, namely LSGP-USFNet, is developed for the efficient detection of ADHD from EEG signals. The proposed method comprised signal segmentation, time–frequency image construction, patch division, Ulam’s spiral and Sophie Germain’s primes-based feature extraction, feature saliency, and classification. A publicly available dataset was used to develop the proposed LSGP-USFNet model. Various patch sizes were used to obtain the highest performance. As given in Table 3, the best average accuracy scores for Part 1 data, Part 2 data, and Part 1 + Part 2 data were 98.97 ± 0.0084%, 98.19 ± 0.0040%, and 97.46 ± 0.0104%, respectively. The obtained results were compared with the recent state-of-the-art methods proposed using the same dataset. For example, the auto-regressive model coefficients were assessed by Kaur et al. [36] using the Yule–Walker, covariance, and Burg techniques. They employed spectral power and classified with an accuracy of 85% using a k-NN classifier. Several nonlinear characteristics, such as Petrosian, Katz, Higuchi fractal dimension, entropy, and Lyapunov exponent, were extracted from EEG signals by Mohammadi et al. [37]. They reported accuracies of 93.65% and 92.28% for relevant features chosen using minimal redundancy maximum relevance and double input symmetrical relevance approaches, respectively, with the neural network (NN) classifier. Moghaddari et al. [11] used a 13-layered convolutional neural network (CNN) model to detect ADHD from EEG signals. The authors initially extracted the rhythms from the EEG signals and converted them into time–frequency images, and these images were used as input to the developed 13-layered convolutional neural network model. The developed model produced an accuracy of 98.48%. Chang et al. [38] extracted deep features from a pre-trained CNN model and classified them with a LSTM classifier for EEG-based ADHD detection. The authors obtained an 85.5% accuracy score. Maniruzzaman et al. [13] used hybrid features from the time and frequency domains. They used a feature selection algorithm to detect the more efficient features. Their method detected ADHD with an accuracy of 97.5% using a Gaussian process-based classifier. Alim et al. [16] extracted time and frequency domain features and feature saliency with the PCA method. Their method obtained an accuracy of 94.2% using the SVM classifier. Güven et al. [17] used Lempel–Ziv complexity and fractal-dimension functions for feature extraction from EEG signals. Their method reported an accuracy of 93.2% in detecting ADHD using a Naïve Bayesian classifier. In Table 10, a comparison of the proposed method with some of the recent works on the same dataset is given.

It can be noted from Table 10 that Kaur et al. [36], Chang et al. [38], and Mohammadi et al. [37] obtained 85.0%, 85.5%, and 93.7% accuracy scores, respectively, using 30 ADHD and 30 normal children. And our proposed method obtained 98.9% accuracy using the same dataset. Moghaddari et al. [11] obtained an accuracy of 98.48% using the proposed method and reported an improved accuracy score of 98.5% with 31 ADHD and 30 normal children. Alim et al. [16] and Güven et al. [17] reported 94.2% and 93.2% accuracy scores, respectively, using 61 ADHD and 60 normal children. Our proposed method and the Maniruzzaman et al. [13] method reported an accuracy of 97.5% using the same dataset. Khare et al. [8] used a variational mode decomposition and Hilbert transform-based approach for ADHD classification. The authors reported an accuracy of 99.75% using extreme learning machines (ELMs) in the classification stage of the proposed work. But they did not use all the samples obtained from the subject in their experimental study.

The advantages of the proposed method are given below:Using a variable window size in patch extraction enabled the proposed approach to be applicable to various input image sizes.All channels of the EEG signals were used in this study.The location of Sophie Germain’s primes on Ulam’s spiral produced a unique pattern for the images.Feature selection reduced the number of gray tones, significantly reducing the run time of classifiers within acceptable limits.By using the patch extraction on the whole time–frequency input image, the need for rhythm extraction was eliminated. The rhythm-based features were also acquired with the patch extraction procedure.

The disadvantages of the proposed method are as follows:The ReliefF feature selection method is time-consuming.The rotational invariance property of the proposed method was not fully investigated.We have used only 61 ADHD and 60 normal children in this work.

In the future, we plan to validate our developed model using a huge dataset obtained from many centers from various countries to make the model more generalizable. Also, to develop trust in our generated model, we propose to employ explainable artificial intelligence (XAI) by visualizing the locations of those responsible for high classification performance [39,40]. These developed models are prone to noise when employed in real-world scenarios. To evaluate the influence of noise on the developed model or due to data, uncertainty quantification (UQ) can be used [41]. Bayesian networks, Monte Carlo dropout, and Fuzzy methods can be employed to make accurate decisions [42].

Also, in the future, we plan to use electrocardiogram (ECG) signals for the automated detection of ADHD [43] and also explore the possibility of using multi-modal signals for the automated detection of children’s mental health [44].

## 5. Conclusions

In this paper, a novel model, LSGP-USFNet, has been developed for the accurate detection of ADHD using EEG signals. In the proposed method, all EEG channels and rhythms are used based on the time–frequency image representation. The unique patterns based on patch extraction and the location of Sophie Germain’s primes on the Ulam’s spiral are acquired, and then a feature saliency procedure is employed to obtain the most efficient gray tones from the time–frequency image. The SVM classifier with a ten-fold cross-validation was used to detect ADHD accurately (97.46%). The limitation of this work is that we have used only one public dataset to develop the proposed model. In the future, we plan to validate our automated system with (i) a huge database with children from different races, (ii) polar Ulam’s spiral to improve the performance, and (iii) a model to detect other neurological disorders of the children. In the future, we intend to use a large diverse database encompassing different races. Additionally, we aim to explore other spiral patterns to complement our current findings. Also, our model can be used to detect other neurological disorders, thereby demonstrating the effectiveness and efficiency of the proposed methodology.

## Figures and Tables

**Figure 1 sensors-23-07032-f001:**
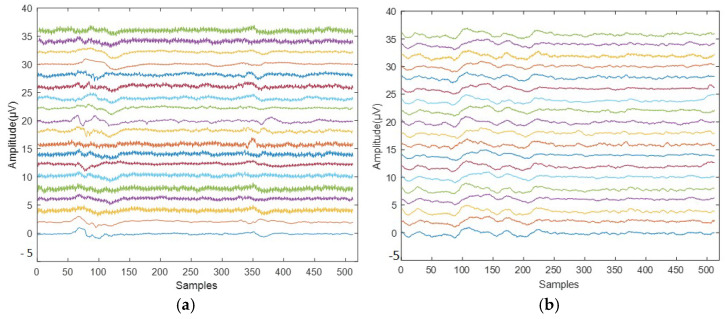
Sample EEG signals: (**a**) ADHD class and (**b**) normal class.

**Figure 2 sensors-23-07032-f002:**
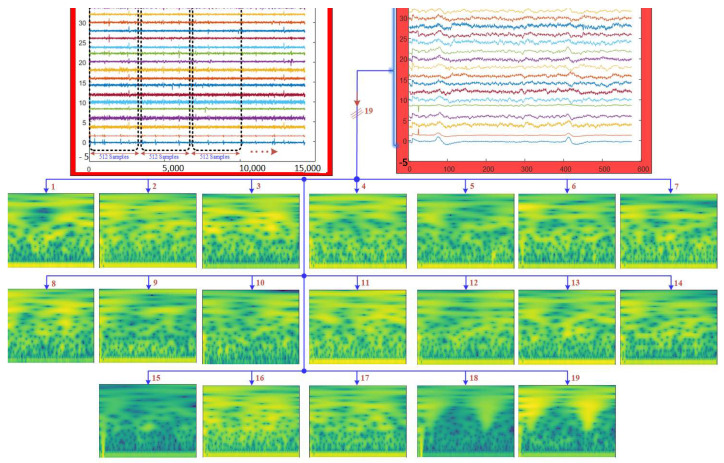
Illustration of EEG signal segmentation and time–frequency representation using the CWT.

**Figure 3 sensors-23-07032-f003:**
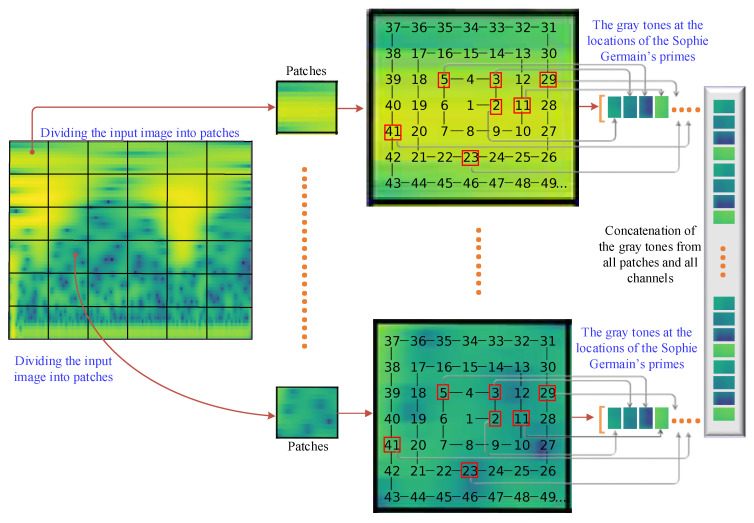
Non-overlapping patch division and feature extractions based on the locations of Sophie Germain’s primes on Ulam’s Spiral.

**Figure 4 sensors-23-07032-f004:**
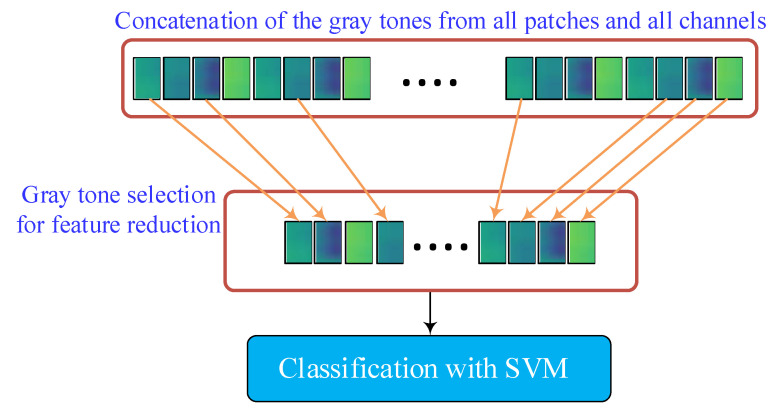
Feature reduction by feature selection and feature classification.

**Figure 5 sensors-23-07032-f005:**
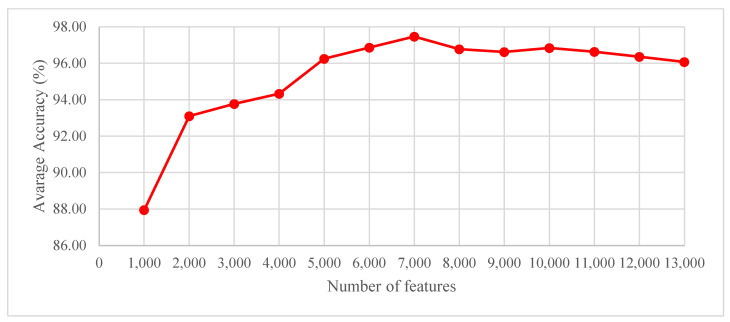
Graph of average accuracy scores (%) versus the number of features for the Part 1 dataset.

**Table 1 sensors-23-07032-t001:** Summary of the works done automated ADHD detection using EEG signals.

Authors	Method	Dataset	Class	Accuracy (%)	Limitations
Khare et al. [6]	Variational mode decomposition (VMD) and Hilbert Transform (HT)-based feature extraction and machine-learning-based classification	IEEE dataport	2	99.90	Highly complex and missing data samples
Bakhtyari et al. [7]	Convolutional long short-term memory and attention mechanisms	Own dataset	2	99.34	Highly complex and intensive computational load
Khare et al. [8]	VMD-HT-based statistical features and extreme learning machines	IEEE dataport	2	99.95	Highly complex and missing data samples
Cehn et al. [9]	Channel selection and deep learning	Own dataset	2	94.67	Highly complex and low accuracy
Dubreuil-Vall et al. [10]	Time–frequency images and convolutional neural network (CNN)	Own dataset	2	88.00	Low accuracy
Moghaddari et al. [11]	Rhythm extraction and CNN	Own dataset	2	98.48	Highly complex and intensive computational load
Ghaderyan et al. [12]	Dynamic frequency warping (DFW) on rhythms of EEG signals and SVM	Own dataset	2	99.17	Highly complex and intensive computational load
Maniruzzaman et al. [13]	Hybrid channel selection, fractal and nonlinear statistical features, and SVM	IEEE dataport	2	97.5	Highly complex and intensive computational load
Vahid et al. [14]	Deep-learning-based EEGNet model and LOOS validation method	Own dataset	2	82.00	Low accuracy
Tor et al. [15]	Empirical mode decomposition (EMD) and discrete wavelet transform (DWT)-based features and feature selection and K-nearest neighborhood (KNN) classifier	Own dataset	3	97.88	Highly complex and intensive computational load
Alim et al. [16]	Statistical properties, time domain and frequency domain features, and SVM	IEEE dataport	2	94.2	Highly complex and intensive computational load
Güven et al. [17]	Lempel–Ziv complexity and fractal-dimension-based feature extraction and Naive Bayes (NB) classifier	Own dataset	2	79.54	Low accuracy

**Table 2 sensors-23-07032-t002:** Performance measures obtained for a patch size of 9 × 9 for Part 1, Part 2, and Part 1 + Part 2.

	Accuracy (%)	Sensitivity (%)	Precision (%)	F1 Score (%)
Part 1	98.78 ± 0.0070	98.82 ± 0.0137	98.24 ± 0.0081	98.53 ± 0.0085
Part 2	98.52 ± 0.0063	98.28 ± 0.0151	98.57 ± 0.0096	98.41 ± 0.0069
Part 1 + Part 2	97.20 ± 0.0058	97.78 ± 0.0093	95.97 ± 0.0120	96.86 ± 0.0064

**Table 3 sensors-23-07032-t003:** Performance measures obtained for a patch size of 15 × 15 for Part 1, Part 2, and Part 1 + Part 2.

	Accuracy (%)	Sensitivity (%)	Precision (%)	F1 Score (%)
Part 1	98.97 ± 0.0084	99.06 ± 0.0122	98.49 ± 0.0088	98.77 ± 0.0101
Part 2	98.19 ± 0.0040	97.53 ± 0.0141	98.11 ± 0.0079	97.87 ± 0.0050
Part 1 + Part 2	97.46 ± 0.0104	96.98 ± 0.0164	97.58 ± 0.0117	97.28 ± 0.0112

**Table 4 sensors-23-07032-t004:** Performance measures obtained for a patch size of 25 × 25 for Part 1, Part 2, and Part 1 + Part 2.

	Accuracy (%)	Sensitivity (%)	Precision (%)	F1 Score (%)
Part 1	98.25 ± 0.0025	98.71 ± 0.0067	98.16 ± 0.0043	98.44 ± 0.0027
Part 2	97.37 ± 0.0158	96.79 ± 0.0220	97.59 ± 0.0184	97.17 ± 0.0170
Part 1 + Part 2	96.09 ± 0.0110	95.55 ± 0.0095	95.63 ± 0.0185	95.58 ± 0.0121

**Table 5 sensors-23-07032-t005:** Performance measures obtained for a patch size of 45 × 45 for Part 1, Part 2, and Part 1 + Part 2.

	Accuracy (%)	Sensitivity (%)	Precision (%)	F1 Score (%)
Part 1	96.14 ± 0.0153	95.27 ± 0.0278	96.23 ± 0.0160	95.84 ± 0.0168
Part 2	97.90 ± 0.0113	96.82 ± 0.0167	98.10 ± 0.0158	97.45 ± 0.0136
Part 1 + Part 2	93.51 ± 0.0125	91.59 ± 0.0259	93.60 ± 0.0163	92.56 ± 0.0149

**Table 6 sensors-23-07032-t006:** Performance measures obtained for a patch size of 75 × 75 for Part 1, Part 2, and Part 1 + Part 2.

	Accuracy (%)	Sensitivity (%)	Precision (%)	F1 Score (%)
Part 1	95.70 ± 0.0134	92.32 ± 0.0339	97.18 ± 0.0161	94.65 ± 0.0179
Part 2	94.08 ± 0.0268	92.78 ± 0.0398	94.52 ± 0.0297	93.60 ± 0.0294
Part 1 + Part 2	91.52 ± 0.0183	88.66 ± 0.0273	91.89 ± 0.0239	90.22 ± 0.0211

**Table 7 sensors-23-07032-t007:** Performance measures obtained for the whole image for Part 1, Part 2, and Part 1 + Part 2.

	Accuracy (%)	Sensitivity (%)	Precision (%)	F1 Score (%)
Part 1	93.26 ± 0.0114	88.21 ± 0.0254	95.26 ± 0.0272	91.55 ± 0.0142
Part 2	92.81 ± 0.0190	90.36 ± 0.0320	94.08 ± 0.0239	92.15 ± 0.0213
Part 1 + Part 2	88.40 ± 0.0185	83.45 ± 0.0388	89.61 ± 0.0199	86.37 ± 0.0236

**Table 8 sensors-23-07032-t008:** Comparison of performance measures obtained using our proposed method with state-of-the-art feature selection algorithms for a patch size of 15 × 15 in Part 1.

	Accuracy (%)	Sensitivity (%)	Precision (%)	F1 Score (%)
ReliefF	98.97 ± 0.0084	99.06 ± 0.0122	98.49 ± 0.0088	98.77 ± 0.0101
NCA	95.46 ± 0.0175	93.39 ± 0.0381	95.61 ± 0.0242	94.43 ± 0.0224
PCC	94.63 ± 0.0232	96.11 ± 0.0232	91.45 ± 0.0282	93.70 ± 0.0266
TVFS	94.38 ± 0.0103	90.57 ± 0.0207	95.67 ± 0.0164	93.03 ± 0.0130

**Table 9 sensors-23-07032-t009:** Running times obtained for feature extraction with various dimensions of the patch window.

	9 × 9	15 × 15	25 × 25	45 × 45	75 × 75
Part 1	1431 s	1095 s	1041 s	1000 s	940 s
Part 2	1260 s	1117 s	1102 s	1049 s	984 s

**Table 10 sensors-23-07032-t010:** Performance comparison of the proposed method with the state-of-the-art techniques using the same dataset.

Study	Data	Features	Classifier	Validation Method	Accuracy (%)
Kaur et al. [36]	ADHD: 30 Normal: 30	Yule–Walker, covariance, and Burg techniques	k-NN	-	85.0
Mohammadi et al. [37]	ADHD: 30 Normal: 30	Fractal dimension, entropy, and Lyapunov exponent	NN	-	93.7
Moghaddari et al. [11]	ADHD: 31 Normal: 30	Deep features	CNN	Cross-validation 5-fold	98.5
Chang et al. [38]	ADHD: 30 Normal: 30	Rhythm features	LSTM	Cross-validation 5-fold	85.5
Maniruzzaman et al. [13]	ADHD: 61 Normal: 60	Hybrid features	Gaussian Process	Cross-validation 5-fold	97.5
Alim et al. [16]	ADHD: 61 Normal: 60	Statistical and time–frequency features	SVM	Cross-validation 5-fold	94.2
Güven et al. [17]	ADHD: 61 Normal: 60	Lempel–Ziv complexity and fractal-dimension	Naive Bayes	LOSO	93.2
Khare et al. [8]	ADHD: 61 Normal: 60	VHERS	ELM	Cross-validation 10-fold	99.7
Proposed method	ADHD: 30 Normal: 30	15 × 15 Ulam’s spiral and Sophia Germain’s primes-based features	SVM	Cross-validation 10-fold	98.9
Proposed method	ADHD: 31 Normal: 30	9 × 9 Ulam’s spiral and Sophia Germain’s primes-based features	SVM	Cross-validation 10-fold	98.5
Proposed method	ADHD: 61 Normal: 60	15 × 15 Ulam’s spiral and Sophia Germain’s primes-based features	SVM	Cross-validation 10-fold	97.5

## Data Availability

Not applicable.

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
