# Peer review of "LSGP-USFNet: Automated Attention Deficit Hyperactivity Disorder Detection Using Locations of Sophie Germain’s Primes on Ulam’s Spiral-Based Features with Electroencephalogram Signals"

_sensors, 2023, doi:10.3390/s23167032_

Round 1

Reviewer 1 Report

The paper addresses an important issue in ADHD detection; however to further improve quality of the paper, the paper requires more detailed explanations of the methodology, stronger justifications for the chosen features, rigorous evaluation against existing methods, and a more comprehensive discussion of the limitations and future directions. A few venues of improvements are as follows:

1. The methodology section should ensure reproducibility. The authors should provide more clarity in the steps involved in signal filtering, segmentation, and time-frequency analysis using continuous wavelet transform so that the reader could easily validate the methods and reproduce the results. 

2. While the utilization of Ulam's spiral and Sophie Germain's primes is intriguing, the authors should provide a stronger justification for their choice and explain how these features are specifically relevant to ADHD detection. Without a clear theoretical foundation, the significance of these patterns remains unclear.

3. The use of ReliefF for gray tone selection is a notable approach, but the authors need to provide more information about the algorithm's parameters and how it determines the most important features for ADHD detection. Additionally, it would be beneficial to compare the results of ReliefF with other feature selection techniques to demonstrate its effectiveness.

4. The paper lacks a thorough evaluation of the proposed method against existing approaches in the literature. It is crucial to compare the performance of LSGP-USFNet with other state-of-the-art methods to demonstrate its superiority. Without this comparison, it is challenging to determine the true effectiveness of the proposed model.

5. The authors briefly mention future work but fail to provide concrete plans for addressing the limitations and potential improvements. It would be beneficial for the authors to outline a specific roadmap for further research, including plans to validate the model with larger and more diverse databases, explore alternative spiral patterns, and extend the model's applicability to other neurological disorders.

6. The paper would benefit from a stronger discussion section that critically analyzes the results and addresses potential confounding factors or sources of error. Additionally, the authors should discuss the clinical implications of their findings and how the proposed LSGP-USFNet model could be integrated into existing diagnostic practices.

Minor English corrections needed.

Author Response

Thank you. Please see attached.

Reviewer 2 Report

Authors proposed LSGP-USFNet for EEG signal processing work. Overall, the manuscript is well written.  There are no English grammar issues. The format of the manuscript seems to be wrong.

There are some comments below.

1. Please use abbreviated journal name in Reference section. Please see author guidelines of MDPI.

2. Informed Consent Statements / Conflicts of Interest need to be mentioned.

3. Please add ref. (In classification, SVM aims to find an~) with ref. (https://www.mdpi.com/1424-8220/22/15/5860).

4. No data availability section.  Please see the author guidelines of MDPI more carefully.

5. The font of the manuscript is something wrong. Please see author guidelines of MDPI.

6. Figures 1,2, and 3 fonts are small.

7. Please change Fig. to Figure.

8. What is the unit of the Amplitude in Figure 1a) ?

9. What is the y-axis label in Figure 2 ?

10. Why authors choose a 512-sample length ?

Author Response

Thank you. Please see attached.

Round 2

Reviewer 1 Report

All the comments raised are addressed.